# Treatment of Osteoporosis in Men on Androgen Deprivation Therapy in Japan

**DOI:** 10.3390/medicina60040551

**Published:** 2024-03-28

**Authors:** Hanako Nishimoto, Atsuyuki Inui, Yutaka Mifune, Kohei Yamaura, Yukari Bando, Yasuyoshi Okamura, Takuto Hara, Tomoaki Terakawa, Ryosuke Kuroda

**Affiliations:** 1Department of Orthopaedic Surgery, Kobe University Graduate School of Medicine, Kobe 650-0017, Japan; hanakoni@med.kobe-u.ac.jp (H.N.); m-ship@kf7.so-net.ne.jp (Y.M.); kurodar@med.kobe-u.ac.jp (R.K.); 2Department of Urology, Kobe University Graduate School of Medicine, Kobe 650-0017, Japan; yukaribando@hotmail.co.jp (Y.B.); dfgky008@gmail.com (Y.O.); supermarimo85@gmail.com (T.H.);

**Keywords:** osteoporosis, androgen deprivation therapy, treatment

## Abstract

*Background and Objectives*: Androgen deprivation therapy (ADT) for prostate cancer has greatly improved treatment outcomes. As patient survival rates have increased, reports of decreased bone density and increased bone fractures as side effects of ADT have emerged. The prevalence of osteoporosis in Japanese men was 4.6%. The purpose of this study was to evaluate the effect of osteoporosis treatment in prostate cancer patients who underwent ADT in Japan. *Materials and Methods*: The subjects were 33 male patients who had undergone ADT for prostate cancer, who were noted to have decreased bone density. Mean age was 76.2 ± 7.7 years (64–87). Medications included vitamin D in one case, bisphosphonates (BP) in 27 cases, and denosumab in five cases. The evaluation method examined the rate of change in bone mineral density (BMD) before osteoporosis treatment and 1 year after. For comparison, a group without osteoporosis treatment intervention (*n* = 33) was selected, and matched for prostate cancer treatment and age. The rate of change in trabecular bone score (TBS) was also calculated. *Results*: The percentage changes in BMD before and 1 year after treatment were as follows: lumbar spine, 7.1 ± 5.8% in the treatment group versus −3.9 ± 4.1% in the no treatment group; femoral neck, 5.5 ± 6.2% in the treatment group versus −0.9 ± 3.9% in the no treatment group; total femur, 6.6 ± 6.4% in the treatment group versus the no treatment group which was −1.7 ± 3.2%. In all cases, there was a clear significant difference (*p* < 0.01). The percent change in TBS was further calculated in the same manner. There was no significant difference between the two groups: +1.7 ± 3.8% in the treated group versus +0.3 ± 4.1% in the untreated group. *Conclusions*: Osteoporosis treatment in Japanese patients with prostate cancer on ADT therapy was found to significantly increase BMD compared to the untreated group. BP and denosumab were found to be very effective in increasing BMD.

## 1. Introduction

Androgen deprivation therapy (ADT) is increasingly used in the treatment of prostate cancer. ADT has traditionally been a treatment reserved for advanced prostate cancer. However, over the past few decades, its use as a neoadjuvant and adjuvant has increased especially for older patients [1,2]. Because androgens are essential for the physiological activity of various body functions, possible side effects of ADT include decreased libido, erectile dysfunction, fatigue, hot flashes, changes in body composition, atherosclerosis, new onset diabetes, cognitive decline, and osteoporosis and subsequent fractures [3]. Among the patients receiving ADT, the overall incidence of fractures has been reported to be significantly higher in the ADT group (19.4%) than in the non-ADT group (12.6%) at 1–5 years after diagnosis [4]. Berruti et al. studied 35 prostate cancer patients without bone metastases who had received luteinizing hormone-releasing hormone analogs for 12 months. A bone mineral density (BMD) loss of 2% or more was observed in 19 patients (54.3%) at the lumbar spine and 15 (42.9%) at the hip joint [5]. Morote et al. reported on the duration of ADT treatment and the incidence of osteoporosis. The incidence of osteoporosis was 35.4% with no ADT treatment, 42.9% at 2 years of ADT, 49.2% at 4 years, 59.5% at 6 years, 65.7% at 8 years, and 80.6% at 10 years or more [6]. This report suggests that the rate of osteoporosis increases with increasing duration of ADT. In Japan, a manual for the management of bone loss due to cancer treatment (cancer treatment-induced bone loss, CTIBL) has been proposed in 2020 [7]. The manual recommends drug treatment for patients with a BMD T-score of −2.0~−1.5 and a family history of hip fracture or a 10-year probability of major osteoporotic fracture by FRAX^®^ of 15% or greater, or a BMD T-score of less than −2.0. The clear establishment of these manuals suggests that sex hormone withdrawal therapy, due to prostate and breast cancer, will become more important in Japan.

Increasing survival rates for prostate cancer have led to an increase in the number of patients undergoing long-term ADT [8]. In Japan, the prevalence of osteoporosis in men is 4.6% [9], which is less than the 6.0% prevalence of osteoporosis in men in Europe [10]. Compared to the prevalence of osteoporosis in the United States, the prevalence is higher in Japan among men in their 80s and older, while the prevalence is lower among Japanese men in their 50s–70s [11]. As for women, the prevalence of osteoporosis is known to be higher than in other countries, but a large cross-sectional study reported that the overall prevalence of osteoporosis in Japan has been decreasing in recent years. The reasons for this are an increase in oral calcium intake, the active awareness-raising activities of the government and academic societies, and an increased awareness of bone health care shown among the participants in a cross-sectional study [9]. Despite the growing understanding of treatment for osteoporosis, there have been no reports in Japan on the effectiveness of treatment for osteoporosis in men undergoing ADT treatment for prostate cancer. The aim of this study is to reveal the effectiveness of osteoporosis treatment in men undergoing ADT treatment for prostate cancer.

## 2. Materials and Methods

### 2.1. Ethical Approval

This study was approved by the ethics committee of Kobe University (IRB No. B190287).

### 2.2. Data Collection

All patients in the study were patients who had visited a urologist and had been diagnosed with prostate cancer. These patients’ data were retrospectively collected from December 2019 to November 2023. They were indicated for ADT for prostate cancer and underwent bone mineral density (BMD) measurement. BMD was measured with a dual-energy X-ray absorptiometry (DXA) device using Horizon A (Hologic, Inc., Marlborough, MA, USA). As defined by CTIBL, patients with a BMD T-score of less than −2.0 were defined as patients with decreased BMD who needed treatment.

The subjects were 33 male patients with prostate cancer, who were found to have decreased BMD. The mean age was 76.2 ± 7.7 (62–88) years. The mean time from the start of ADT to the start of osteoporosis treatment was 3.9 (0–11) years. The drugs used for treatment included vitamin D in one patient, bisphosphonates (BP) in 27 patients, and denosumab in five patients. The basic policy for medication is based on the T-score: denosumab is given to patients with a T-score of less than −3.3 SD, and BP is given to patients with a T-score greater than −3.3 SD. There was one case who underwent gastrectomy before the prostate cancer treatment. For this patient, VitD was the sole dose case because of chronic Ca malabsorption and low serum 25(OH)D levels. These patients had no history of medical conditions leading to secondary osteoporosis (hyperparathyroidism, hyperthyroidism, anorexia nervosa, diabetes mellitus, rheumatoid arthritis, chronic kidney disease, chronic obstructive pulmonary disease, and heavy alcohol drinking). The breakdown of ADT therapy for prostate cancer was Leuprorelin Acetate in six cases, Leuprorelin Acetate and Bicalutamide in 24 cases, and Degarelix Acetate and Bicalutamide in three cases. Radium therapy was not used in these patients. All of these patients also had adequate vitamin D intake. They were verbally advised prior to the observation period to consume at least 600 mg of calcium per day through their diet.

An age-matched no treatment intervention group for osteoporosis (*n* = 33) was selected for comparison with these groups with osteoporosis treatment intervention. The no intervention group was receiving ADT for prostate cancer but their BMD did not decrease during the study period. The background of these patients is shown in Table 1. Significant differences were found only in serum 25(OH)D levels.

The rate of change in BMD at initial treatment and 1 year after treatment was evaluated for both groups. The calculation method of the rate of change is as follows:(BMD at 1 year − BMD at initial examination)/BMD at initial examination × 100

Statistics were performed using the paired *t*-test. For the BMD measurement, lumbar spine, femoral neck, and total hip were used based on osteoporosis treatment guideline and CTIBL. For the osteoporosis treatment intervention group, the percentage change in BMD was analyzed by the type of drug treatment. For these groups, we additionally evaluated the trabecular bone score (TBS), which is said to reflect the trabecular bone structure. TBS is an index calculated by textural analysis of pixel density in lumbar spine images measured by a DXA device. TBS and BMD are analyzed based on the same scan data, but their calculation methods are different; BMD is related to the total brightness of each pixel in a bone image but does not consider the variation in values between pixels. On the other hand, the TBS algorithm analyzes the spatial variation in pixel luminance. Specifically, it analyzes the spatial variation of pixel luminance by determining the difference between the luminances of adjacent pixels and calculating the square of the difference. Therefore, TBS is considered to correlate with bone microstructure indices, such as trabecular number, trabecular separation, and connectivity density [12]. In this study, TBS was measured using TBS iNsight™ (Medimaps, Basel, Switzerland). There are also no reports of changes in TBS values in response to treatment. We evaluated the TBS score by the rate of change between the initial treatment and one year after treatment.

The calculation method for the percentage change in TBS is as follows:(TBS after 1 year − TBS at initial visit)/TBS at initial visit × 100.

## 3. Results

### 3.1. Change in BMD

BMD and TBS values at the initial treatment and one year after the treatment are shown in Table 2.

The BMD of the treated group changed from 0.799 ± 0.169 before treatment to 0.845 ± 0.171 after treatment in the lumbar spine, from 0.555 ± 0.066 before treatment to 0.564 ± 0.067 after treatment in the femoral neck, and from 0.695 ± 0.084 before treatment to 0.707 ± 0.086 after treatment in the proximal femur. BMD in the untreated group changed from 1.025 ± 0.170 before to 0.982 ± 0.162 after treatment in the lumbar spine, from 0.693 ± 0.087 before to 0.683 ± 0.081 after treatment in the femoral neck, and from 0.878 ± 0.092 before to 0.859 ± 0.084 after treatment in the proximal femur.

The percentage change in BMD before and one year after treatment was 7.1 ± 5.8% in the treated group and −3.9 ± 4.1% in the untreated group for the lumbar spine (Figure 1a), 5.5 ± 6.2% in the treated group and −0.9 ± 3.9% in the untreated group for the femoral neck (Figure 1b), and 6.6 ± 6.4% in the treated group and −1.7 ± 3.2% in the untreated group for the proximal femur (Figure 1c). BMD change in the intervention group was significantly higher than in untreated group (*p* < 0.01).

### 3.2. Change in TBS

TBS at the initial treatment and one year after the treatment are shown in Table 1. TBS in the treated group changed from 1.242 ± 0.074 before treatment to 1.263 ± 0.080 after treatment. On the other hand, in the untreated group, it was 1.310 ± 0.063 before treatment and 1.313 ± 0.072 after treatment. The change in TBS was +1.7 ± 3.8% in the treated group versus +0.3 ± 4.1% in the untreated group, with no statistically significant difference between the two groups (Figure 2).

### 3.3. Change in BMD and TBS by Drug Type

BMD and TBS values at the initial treatment and one year after the treatment by drug type are summarized in Table 3.

BMD in the vitamin D group changed from 0.759 before to 0.778 after treatment in the lumbar spine, from 0.522 before to 0.543 after treatment in the femoral neck, and from 0.648 before to 0.655 after treatment in the proximal femur. BMD in the BP treatment group changed from 0.568 ± 0.064 before treatment to 0.577 ± 0.064 after treatment in the lumbar spine, from 0.568 ± 0.064 before treatment to 0.577 ± 0.064 after treatment in the femoral neck, and from 0.712 ± 0.078 before treatment to 0.724 ± 0.078 after treatment in the proximal femur. TBS was 1.253 ± 0.070 before treatment and 1.274 ± 0.079 after treatment. BMD in the denosumab-treated group increased from 0.644 ± 0.179 before treatment to 0.740 ± 0.183 after treatment in the lumbar spine, from 0.492 ± 0.036 before treatment to 0.493 ± 0.031 after treatment in the femoral neck, and from 0.614 ± 0.067 before treatment to 0.621 ± 0.077 after treatment in the proximal femur. TBS was 1.181 ± 0.064 before treatment and 1.199 ± 0.053 after treatment.

At the lumbar spine, the BMD increase was 6.9% in the VitD group, 6.3% in the BP group, and 11.6% in the denosumab group (Figure 3a). In the femoral neck, this was 3.0% in the VitD group, 5.5% in the BP group, and 5.6% in the denosumab group (Figure 3b). BMD of proximal femur increased by 2.7% in the VitD group, 6.6% in the BP group, and 7.2% in the denosumab group (Figure 3c). There were no cases of fracture during the study period.

At the lumbar spine, the TBS increase was 2.3% in the VitD group, 1.7% in the BP group, and 1.6% in the denosumab group (Figure 4).

## 4. Discussion

Androgen deprivation therapy (ADT) has been used in many cases for metastatic prostate cancer. As a result, an increasing number of patients are receiving long-term ADT therapy [13]. Various side effects have been reported for ADT. ADT results in a rapid decrease in testosterone. This rapid decrease in testosterone can lead to side effects such as vasomotor flushing, fatigue, sexual dysfunction, skeletal-related events, anemia, metabolic/cardiovascular, and cognitive dysfunction [14]. In particular, there are many reports that long-term ADT may cause a decrease in bone mineral density (BMD) [15]. Although BMD reduction is often asymptomatic, it has been reported that up to 20% of men undergoing ADT may eventually experience a fracture. In a report examining the records of 50,613 men diagnosed with prostate cancer between 1992 and 1997, 19.4% of those who received ADT had a fracture, compared with 12.6% of those who did not receive androgen deprivation therapy (*p* < 0.001) [4]. A report on the prevalence of osteoporosis-related conditions at the time of ADT induction showed a prevalence of 4.3% for osteoporosis and 35.7% for osteopenia in 115 men (mean age 73.3 ± 7.6 years) with ADT induction [16]. FRAX calculations revealed a 10-year fracture risk incidence of 4.4% for osteoporotic fractures and a 10-year average fracture risk incidence of 1.7% for femoral neck fractures. Spinal X-ray imaging revealed at least one vertebral fracture in 32.2% cases. Bone loss during ADT treatment has been reported to be greatest in the first year after the start of ADT. Decreases of −2.5% in the total femur, −2.4% in the proximal femur, −2.6% in the radius, and −4.0% in the lumbar spine occurred during the first year after the start of ADT treatment [15]. These reports indicate that about 30% of prostate cancer patients treated with ADT may have osteoporosis, and that the first year after treatment is the period that requires the greatest attention.

Therefore, it is recommended to measure baseline BMD using DXA prior to the start of ADT, as well as periodic BMD measurements based on the initial T-score [13].

Furthermore, fractures during ADT therapy have been reported. During a mean follow-up period of 47.7 months, 977 patients (3.43%) developed osteoporosis fracture (OF), and the incidence of hip, spine, and wrist fractures differed significantly between the ADT and non-ADT groups [17]. The incidence of OF was significantly higher in the ADT group than in the non-ADT group, and the incidence of spine, hip, and wrist fractures was significantly higher in the ADT group than in the non-ADT group, regardless of the stage of prostate cancer. Fractures during ADT treatment have been reported to shorten survival. Survival rate tended to decrease in the group with a history of pathological fracture compared to the group without pathological fracture [11]. Median survival period was 121 and 160 months in men without and with a history of skeletal fracture since the diagnosis of prostate cancer, respectively [18]. Therefore, it is essential to evaluate BMD using DXA in patients undergoing ADT treatment to improve life outcomes. In addition, therapeutic intervention is required when bone loss is diagnosed [18]. There are several reports regarding the treatment of osteoporosis during ADT therapy.

Lifestyle modifications such as increased exercise, calcium (1500 mg) and vitamin D (800 IU) supplementation, smoking cessation, decreased alcohol consumption, and weight loss are suggested therapies to prevent fractures [13].

There are numerous reports of drug therapy for osteoporosis during ADT treatment. Through subgroup analyses of the present study, both zoledronic acid and alendronate showed a significant improvement in BMD percentage changes. Diphosphonates significantly increased BMD percentage changes of the lumbar spine, total hip, and femoral neck in men receiving androgen deprivation therapy for prostate cancer. BP is effective in preventing BMD decrease in men undergoing ADT [13]. Wu et al. reported that the use of BP in 920 patients undergoing ADT significantly improved the rate of BMD change at all sites: lumbar spine, femoral neck, and total hip [19]. In the group that received alendronic acid orally once a week for one year, BMD increased by 3.7% in the spine and 1.6% in the femoral neck. In contrast, the placebo group showed a decrease of 1.4% in the spine and 0.7% in the femoral neck [20]. Bruder JM et al. compared the alendronic acid-treated and non-treated groups and found that the percentage change in BMD per year was −1.29% versus +1.41% in the lumbar spine, −2.17% versus +0.32% in the femoral neck, −0.94% versus +0.94% for the entire femur, −0.94% vs. +0.97%, respectively, showing statistically significant difference [21].

Risedronic acid is also effective in treatment. A total of 61 prostate cancer patients receiving ADT therapy treated with risedronic acid showed little change in radius and femur BMD while BMD of lumbar spine showed a 4.9% increase [22]. Zoledronic acid has also been reported to be effective against bone mineral loss. A study of 106 men with prostate cancer who received ADT with or without 4 mg of zoledronic acid reported a 5.6% increase in lumbar spine BMD in the treatment group and a 2.2% decrease in the placebo group [23]. The increase in BMD with BP in the present study was 6.3% in the lumbar spine, 5.5% in the femoral neck, and 6.6% in the proximal femur, which compares favorably to previous reports.

Several reports on denosumab administration have also been published. Smith et al. treated 734 male patients receiving ADT therapy with denosumab and compared them to a placebo group. Twenty-four months after initiation of treatment, lumbar spine BMD increased by 5.6% in the denosumab group, compared with a 1.0% decrease in the placebo group. A significant difference between the two groups was observed 1 month after initiation of treatment and persisted through to 36 months. Denosumab therapy also resulted in significant increases in BMD at all sites: the proximal femur, the femoral neck, and the distal third of the radius. The study also showed that denosumab-treated patients had a decreased incidence of new vertebral fractures at 36 months [24]. Although the duration of denosumab treatment in this study was one year, the BMD increase was 11.6% in the lumbar spine, 5.6% in the femoral neck, and 7.2% in the entire proximal femur, which is better than previous reports.

There have been reports of efficacy for selective estrogen receptor modulators as well [25]. A total of 646 men undergoing ADT for prostate cancer were assigned to toremifene (80 mg orally daily) and 638 to placebo, and subjects were followed for 2 years. The 2-year incidence of new vertebral fractures was 4.9% in the placebo group versus 2.5% in the toremifene group. Treatment with toremifene significantly increased BMD in the lumbar spine, hip, and femoral neck compared to the placebo.

Men with non-metastatic prostate cancer (*n* = 48) receiving GnRH agonists were randomly assigned to receive raloxifene (60 mg/day) for 12 months or no raloxifene [26]. Mean BMD of the lumbar spine increased 1.0 ± 0.9% in men who received raloxifene while it decreased 1.0 ± 0.6% in men who did not receive raloxifene. BMD of the proximal femur increased 1.1 ± 0.4% in men who received raloxifene and decreased 2.6 ± 0.7% in men who did not receive raloxifene. Spinal BMD also increased with raloxifene administration.

There have been several reports of elevated markers of bone resorption and bone formation in prostate cancer patients compared to healthy controls [15]. This may be the result of hypermetabolic turnover bone metabolism occurring in patients undergoing ADT treatment, resulting in decreased BMD and increased fracture rates. We speculate that suppression of bone resorption may be effective in androgen deprivation osteoporosis.

In this study, patients undergoing ADT therapy were also evaluated using TBS, which is reported to be an index reflecting the trabecular bone trabecular structure. When the BMD values are the same, the trabecular bone with higher TBS has a more uniform quality [12]. In the present study, the TBS value showed an increase of +1.7% for the treated group. On the other hand, the untreated group showed an increase of +0.3% despite a decrease in BMD. The dissociation between TBS and BMD has been reported in a Chinese population study. In the male group aged 36–85 years, dissociation between TBS and BMD was observed. Generally, males show the highest lumbar spine TBS around the age of 50 years, which is later than the peak age of BMD. This may reflect a time lag between the bone microstructural changes and bone mineral deposition [27]. In the present study, we speculate that the changes in the TBS may have been difficult to capture because of the small absolute value of the TBS and short follow-up period. It has been reported that both TBS and BMD are useful for predicting fractures [28], and the combination of TBS and BMD may be useful for predicting fractures in patients undergoing ADT therapy for prostate cancer.

There are limitations to this study. First, because this is a retrospective study, the number of cases is small. However, as mentioned above, this is due to the small number of Japanese male osteoporosis patients, which makes it difficult to find patients who are suitable for treatment when judged by the T-score. In addition, due to the small number of cases, the number of patients treated with denosumab and vitamin D alone was small, resulting in a bias in the number of cases per group. We intend to collaborate with other institutions in the future to increase the number of cases and prove that the results of this study were accurate. However, we believe that we were able to present a report that shows that bone resorption inhibitors are effective for Japanese men with osteoporosis due to ADT.

As has been reported in the past, it is important to provide osteoporosis management and treatment to prostate cancer patients undergoing ADT therapy. This is the first report of an osteoporosis treatment intervention in Japanese patients undergoing ADT therapy. The intervention group showed a significant increase in BMD compared to the no intervention group. During the observation period of this study, there was not a single case of fracture as an adverse event, and no case resulted in reflux esophagitis, osteonecrosis of the jaw, or atypical femur fracture, which are considered to be serious side effects of bone resorption inhibitors. As mentioned earlier, the incidence of osteopenia and osteoporosis patients correlates with the duration of ADT treatment. There was a −3.8% loss of BMD in the lumbar spine, −0.9% in the femoral neck, and −1.7% in the proximal femur during the first year of ADT therapy. Since these BMD changes were larger than age-related bone mineral density loss, it is important to evaluate BMD periodically in patients receiving ADT treatment. In addition, bone resorption inhibitors have been shown to be very effective against bone loss and should be recommended for patients undergoing ADT.

## 5. Conclusions

Osteoporosis treatment in Japanese patients with prostate cancer on ADT therapy was found to significantly increase BMD compared to the untreated group.

The group with a T-score of less than −3.3 received denosumab and the group with a T-score of −3.3 or greater received BP, resulting in improved BMD.

## Figures and Tables

**Figure 1 medicina-60-00551-f001:**
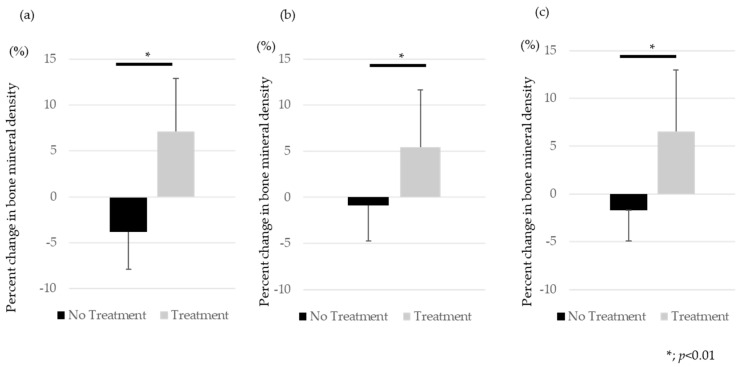
The percentage change in BMD before and one year after treatment. (**a**) Lumber Spine, (**b**) Femoral Neck, (**c**) Total hip.

**Figure 2 medicina-60-00551-f002:**
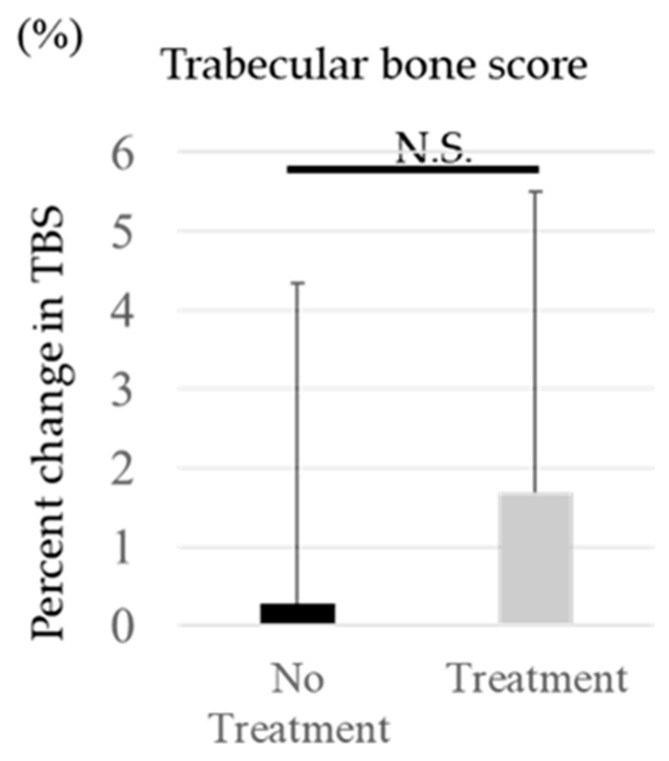
The percentage change in TBS before and one year after.

**Figure 3 medicina-60-00551-f003:**
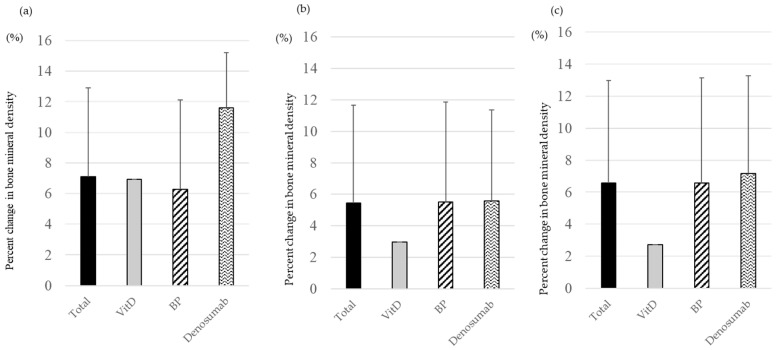
The percentage change in BMD before and one year after by drug type. (**a**) Lumber Spine, (**b**) Femoral Neck, (**c**) Total hip.

**Figure 4 medicina-60-00551-f004:**
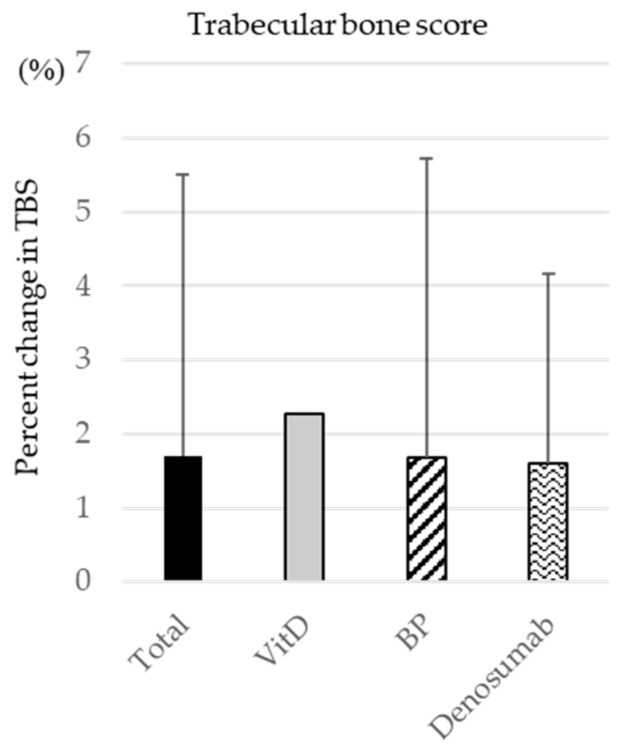
The percentage change in TBS before and one year after by drug type.

**Table 1 medicina-60-00551-t001:** Patients’ backgrounds.

	Treatment (*n* = 33)	No Treatment (*n* = 33)	*p* Value
Age (y.o.)	76.2 ± 7.7	76.0 ± 7.6	0.203
BMI (kg/m^2^)	22.4 ± 3.3	23.4 ± 4.0	0.226
Bone Metabolic Markers			
TRACP 5b	480.3 ± 170.3	457.5 ± 224.0	0.246
P1NP	58.9 ± 16.1	56.1 ± 18.5	0.086
BAP	15.6 ± 7.0	15.1 ± 8.7	0.062
25(OH)D	17.5 ± 6.5	24.3 ± 8.3	0.016 *

* *p* < 0.05.

**Table 2 medicina-60-00551-t002:** BMD and TBS values at the initial treatment and one year after the treatment or initial diagnosis and one year after.

		No Treatment		Treatment	
		Before Follow Up	1 Year Later	Before Treatment	1 Year Later
BMD (g/cm^2^)	Lumber spine	1.025 ± 0.170	0.982 ± 0.162	0.799 ± 0.169	0.845 ± 0.171
	Femoral neck	0.693 ± 0.087	0.683 ± 0.081	0.555 ± 0.066	0.564 ± 0.067
	Total hip	0.878 ± 0.092	0.859 ± 0.084	0.695 ± 0.084	0.707 ± 0.086
Trabecular bone score		1.310 ± 0.063	1.313 ± 0.072	1.242 ± 0.074	1.263 ± 0.080

**Table 3 medicina-60-00551-t003:** BMD and TBS values at the initial treatment and one year after the treatment by drug type.

	Bone Mineral Density (g/cm^2^)						TBS	
	Lumber Spine		Femoral Neck		Total Hip			
	Before Treatment	1 Year Later	Before Treatment	1 Year Later	Before Treatment	1 Year Later	Before Treatment	1 Year Later
Vitamin D	0.759	0.778	0.522	0.543	0.648	0.655	1.272	1.301
Bisphosphonate	0.829 ± 0.150	0.866 ± 0.160	0.568 ± 0.064	0.577 ± 0.064	0.712 ± 0.078	0.724 ± 0.079	1.253 ± 0.070	1.274 ± 0.079
Denosumab	0.644 ± 0.179	0.740 ± 0.183	0.492 ± 0.036	0.493 ± 0.031	0.614 ± 0.067	0.621 ± 0.071	1.181 ± 0.064	1.199 ± 0.053
Total	0.799 ± 0.169	0.845 ± 0.171	0.555 ± 0.066	0.564 ± 0.067	0.695 ± 0.084	0.707 ± 0.086	1.242 ± 0.074	1.263 ± 0.080

## Data Availability

Our data are unavailable due to privacy.

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
