# Peer review of "Treatment of Osteoporosis in Men on Androgen Deprivation Therapy in Japan"

_medicina, 2024, doi:10.3390/medicina60040551_

Round 1

Reviewer 1 Report

Comments and Suggestions for Authors

Summary:

This manuscript reported a study about the effects of anti-osteoporosis medication on the BMD in patients with prostate cancer and undergoing androgen-deprivation therapy. The authors recruited 33 male patients who had undergone ADT for prostate cancer. The mean age was 76.2 ± 7.7 years (64-87). The medications included vitamin D in 1 case, bisphosphonates in 27 cases, and denosumab in 5 cases. Another control group without osteoporosis treatment intervention (n=33) was selected, matched for prostate cancer treatment and age. The results of BMD percent change before and 1 year after treatment was as follows: lumbar spine; 7.1±5.8% in the treatment group versus -3.9±4.1% in control group; femoral neck; 5.5±6.2% in the treatment group versus -0.9±3.9% in control group; total femur; 6.6±6.4% in the treatment group versus -1.7±3.2% in control group. The percent change in TBS showed no significant difference between the two groups: +1.7±3.8% in the treated group versus +0.3±4.1% in the untreated group. The authors concluded that anti-osteoporosis treatment in Japanese patients with prostate cancer on ADT therapy was found to significantly increase BMD compared to the control group. BP and denosumab were found to be very effective in increasing BMD.

Comments:

This manuscript reported a observation study showed that anti-osteoporosis treatment in Japanese patients with prostate cancer on ADT therapy could significantly increase BMD compared to the control group. BP and denosumab were found to be very effective in increasing BMD. This study provided some clinical outcome results in the treatment of low BMD in patients with prostate cancer and on ADT. The results could be expected from previous studies. The results from this study did not provide much new information than that already known. Some more issues needed to addressed to reach a sloid conclusion and enhance its clinical significance. 

Other comments:

1.      The authors included three kinds of anti-osteoporosis medications in this study, vitamin D in 1 case, bisphosphonates in 27 cases, and denosumab in 5 cases. This kind of study design may increase confounding effects because of small numbers in vitamin denosumab group. These patients may not be suitable included.

2.      The authors may need to present the data of T-score for better understanding of the BMD situation. Some patients with high BMD could stand ADT without significant BMD loss. The authors may describe their policy or criteria of anti-osteoporosis treatment for patients with prostate cancer and on ADT.

3.      Did ADT influence the bone markers in this study? Did it influence the fracture rate? The authors may also need to analyze the effects of other co-morbidities.

4.      Did these patients take adequate vitamin D and calcium during study period? How about the dosage?

5.      Lines 22-23. Missing statement about the BMD percentage of total femur should be corrected.

Comments on the Quality of English Language

Comments:

This manuscript reported a observation study showed that anti-osteoporosis treatment in Japanese patients with prostate cancer on ADT therapy could significantly increase BMD compared to the control group. The amnuscript was well-written using English language.

Reviewer 2 Report

Comments and Suggestions for Authors

In this manuscript author evaluated the effect of treatment oteoporosis in patients with prostate cancer undergoing ADT 

Abstract

Correct this sentence from the abstract. Has no context.

The prevalence of osteoporosis in Japanese men 11 was 4.6%

Add BP, BMD meanings

Introduction

Correct adt to ADT in line 40

Shorten the introduction. Data on women is not neccesary

Line 68 and 70 ADL is ADT?

Method

Line 77 please use ADT instead of androgen deprivation therapy as you have introduced

Add the period of patient inclusion.

Why patients were matched only for age and not disease stage? There were differences regarding other variables?

Result

Table showing patients characteristic of intervention and no intervention group is needed

Line 162 what you mean with VITD group, you refer to one patient?

Discussion

Please avoid using too many numbers from other studies.

Line 190 Androgen Deprivation Therapy (ADT), line 200 the same

Androgen Deprivation Therapy (ADT) is the standard of care for metastatic prostate  cancer and has been used in many cases.

I do not agree with this. Please refer to DOI: 10.1016/j.ctarc.2022.100606

Do no repeat information such as the adverse events (already in the introduction)

What about Radium 223 in Japanese?

Please add limitations such as low number of patients, single institution,

Conclusion

Please soften the conclusion as the number of patients is very low

In general

Check all acronism (ADT, etc)

The number of included patients is very low

Reference

Please use a more current reference 1

Comments on the Quality of English Language

moderate correction

Round 2

Reviewer 1 Report

Comments and Suggestions for Authors

This revised manuscript reported an observation study showed that anti-osteoporosis treatment could significantly increase BMD in Japanese patients with prostate cancer on ADT therapy. BP and denosumab were found to be very effective in increasing BMD. This study provided some clinical outcome data in the treatment of low BMD in patients with prostate cancer and on ADT. The revised manuscript addressed the comments sufficiently and provided new information of bone turnover markers. This manuscript in the current status can be considered for publication in Medicina.

Comments on the Quality of English Language

The paper is well-written.